# Improved explainability for quality classifiers demonstrated in ultrasound medical imaging

Mesfin Taye
*School of Electrical and Computer Engineering*
*Oklahoma State University*
Stillwater, OK
mesfin.taye@okstate.edu

Martin Hagan
*School of Electrical and Computer Engineering*
*Oklahoma State University*
Stillwater, OK
mhagan@okstate.edu

*Abstract*—This study introduces a novel refinement to two classes of local explainability techniques and combines this with the Self-Organizing Map (SOM) to achieve a combination of local and global explainability. The new approach is demonstrated on a deep neural network that has been trained to classify the quality of a certain type of ultrasound exam. Using the new approach, we are able to demonstrate that the deep network uses two different types of image characteristics in combination to assess quality. This insight would not have been possible using standard local explainability methods. In a clinical setting, the novel explanations can enhance confidence in the deep network decisions.

*Index Terms*—Explainability, Deep Learning, SOM, FAST

## I. INTRODUCTION

There is a consensus that highly performing Deep Learning (DL) systems are no longer sufficient by themselves. They need to have a mechanism for explaining their decisions. Providing an explanation of the decision making process builds trust and transparency.

It is common to group explainability techniques into two categories: global and local. For global explainability we are interested in the overall pattern or logic of the decision process. For local explainability we want to know what characteristics of a particular input example led to the specific decision made by the DL model. In this paper we combine aspects of both approaches.

Some local explainability techniques use the gradient of the network output with respect to the input elements to define the attributions (explanations), while other methods propagate attributions from the network output back to the input. Integrated Gradient (IG) [1] is an example of the former, and DeepLIFT [2] is an example of the latter. IG computes the average of the gradient along a line. DeepLIFT uses the internal structure of the model to propagate attributions. A commonality between these two methods (as well as other standard methods) is that they both use baselines. Baselines are reference inputs from which we want to measure a change. For example, IG uses baselines to define a path for the line integral, whereas DeepLIFT uses baselines to define the difference-from-reference of the output. Selecting the right baseline for a given problem generally requires domain knowledge. The method of selecting baselines is still an active research area [3].

In image-based problems, it is common to use black images as baselines [1]. In [4], the authors used the expected value of the input as a baseline. Additionally, the averages of multiple baselines and blurred versions of inputs are also used as baselines [4]. In this study, we propose a novel way of selecting baselines. The main idea is that explainability will be improved when the baseline is near the current input but on the opposite side of the decision boundary.

Global explainability focuses more on overall model performance than individual predictions. In [5], the authors calculated the average of all true positive instances from a validation or testing set, weighted by the model's predicted probability of the target label, as a measure of global explainability. Global explainability can also be achieved by extending local explainability methods in some form to make sense at a global level. The authors in [6], for example, used the Local Interpretable model-agnostic explanations (LIME) method to collect a set of instances and form an explanation matrix as a global explainability approach. After calculating the attributes using the Layer-wise Relevance Propagation (LRP) local explainability method, the authors in [7] clustered the attributes to study the global patterns.

In this paper we propose using the Self-Organizing-Map (SOM) [8] as a global explainability method. The SOM is an unsupervised clustering network that enables visualization of high dimensional inputs in two dimensional representations. There are several tools of the SOM that can be used to visualize and interpret Deep Learning models. One advantage of the SOM is that it can be used for high dimensional inputs, and it is global in its nature. The SOM will be used in this study to explain the operation of trained neural networks. We are not aware of any previous research that has used an SOM to explain the operation of a DL model trained with supervised learning.

In the image processing area, most research on explainability has focused on problems of object detection, object classification and image segmentation. There has been much less research on explainability for quality assessment applications, which is a significantly different endeavor and has a more amorphous character. We believe it is an application which requires a combination of local and global explainability techniques. We will use a DL model trained to classify the

quality of Focused Assessment with Sonography in Trauma (FAST) ultrasound exams as a testbed to demonstrate our proposed explainability methods.

In Section II, we present the local explainability method followed by a global explainability method in Section III. We then present the experimental setup in Section IV. Finally, we present the results in Section V.

## II. LOCAL EXPLAINABILITY

In this section we want to begin by giving very brief outlines of two popular local explainability methods (Integrated Gradient [IG] and DeepLIFT [DL]) that are representative of two major categories of local explainability (gradient and attribution-propagation). Both of these methods (IG and DL) use baseline inputs as key components of their algorithms. Next, we will introduce a new approach to selecting the baselines. The new approach can be applied to many other explainability techniques, but we have chosen these two for demonstration purposes.

### A. Integrated Gradient (IG)

In [9], the authors introduced the gradient as a measure of importance for Bayes classifiers. The gradient method measures the change in the network output for a small change in the network input. The higher the gradient, the higher the attribution of the input. There are various flavors of the gradient method.

In this paper we will focus on IG [1], which is the average gradient of the score with respect to the input as it varies along a straight line from a baseline input to the current input. Each point on the line between the baseline and the input is weighted equally. Suppose we have $m$ sample inputs on the line between the baseline and the input. The $t^{th}$ sample input on this line is denoted by $p_t$, for $t \in \{1, 2, ..., m\}$. The IG attribution $_{IG}\mathbf{C}$ of input $\mathbf{P}$ using baseline $\bar{\mathbf{P}}$ is given by

$$\left[ _{IG}\mathbf{C}(\mathbf{P}, \bar{\mathbf{P}}) \right]_{i,j} = \frac{(p_{i,j} - \bar{p}_{i,j})}{m} \sum_{t=0}^{m-1} \frac{\partial n_c^M(\mathbf{P}_t)}{\partial p_{i,j}} \quad (1)$$

where $\mathbf{P}_t = \bar{\mathbf{P}} + \frac{t}{m-1}(\mathbf{P} - \bar{\mathbf{P}})$. See [1] for details of the algorithm.

### B. Deep Learning Important FeaTures(DeepLIFT)

In [10], the authors introduced an attribution-propagation method by defining three constraints (assumptions).

1)
$$c(a_i^m) = \sum_{a_j^{m+1} \in pa(a_i^m)} c(a_i^m | a_j^{m+1}) c(a_j^{m+1}) \quad (2)$$

2)
$$\sum_{a_i^m \in ch(a_j^{m+1})} c(a_i^m | a_j^{m+1}) = 1 \quad (3)$$

3)
$$0 \le c(a_i^m | a_j^{m+1}) \le 1 \quad (4)$$

where $a_i^m$ is the activation of the $i^{th}$ neuron in layer $m$, $ch(a_i^m)$ is the set of all of input neurons of $a_i^m$ (children), $pa(a_i^m)$ is the set of all neurons that $a_i^m$ is an input for (parents), $c(a_i^m)$ is the attribution of neuron $i$ of layer $m$ to the output of the network, and the multiplier $c(a_i^m | a_j^{m+1})$ is the partial contribution of neuron $a_i^m$ to neuron $a_j^{m+1}$,

The first assumption defines the attribution $c(a_i^m)$ as the sum of all the contributions that $a_i^m$ makes to its parent's, $a_j^{m+1}$, attributions. This is to say that a neuron is important to the decision making (output) if it is important to the parent neurons, and the parent neurons are important to the output [10]. The second and the third constraints are used so that attribution is conserved at each layer. The procedure starts by assigning the last layer attribution $c(a_k^M)$ to the output $a_k^M$. It then backpropagates the attribution $c(a_k^M)$ to lower layer neurons using Eq 2. Using Eqs. 2 and 3, it can be shown that

$$\sum_{i=1}^{S^m} c(a_i^m) = a_c^M \quad (5)$$

which is the law of conservation of attribution. This applies to every layer, as well as the input, $a_i^0 = p_i$.

DeepLIFT [2] is an attribution-propagation method where the attributions are computed as a deviation from a baseline input. First, the network response to the baseline input is computed, followed by the response to the current input. At the final layer the network output for class $c$ is

$$n_c^M = \sum_{i=1}^{S^{M-1}} w_{c,i}^M a_i^{M-1} + b_c^M \quad (6)$$

$$a_c^M = f^M \left( n_c^M \right) \quad (7)$$

The deviation from baseline is

$$\Delta a_c^M = a_c^M - \bar{a}_c^M \quad (8)$$

where $\bar{a}_c^M$ is the network output when a baseline input is applied. DeepLIFT then follows a backpropagation scheme modeled on Eq. 2. (See [2] for details.)

### C. Weighted Top 5 Closest Baseline

In this section we propose a new method for selecting baseline inputs. We will demonstrate the procedure for DeepLIFT and IG, but it can be used with any explainability method that uses a baseline input. The main idea behind the new method is that explainability will be improved if the current input and the baseline input span the decision boundary. For a binary classification problem, if the input is from the positive class, then the baseline will be selected as the closest input in the training set that is from the negative class.

First, denote the training set by $T$. Then, split the training set into pairwise disjoint subsets such that each subset contains all the data that belongs to a class. We denote elements of class $i$ by $T_i$ such that $T_i \cap T_j = \emptyset, \quad \forall i \ne j$. For example, $T_1$ represents the set of elements of class 1 in the training set. For this study, we focus on a training set of only two classes.

If we have input $\mathbf{P}$, we denote the $k^{th}$ closest baseline input by $^k\mathbf{P}^*(\mathbf{P})$, which is given by

$$^k\mathbf{P}^*(\mathbf{P}) = \underset{\tilde{\mathbf{P}} \in T_i^c - {}^kU}{argmin}\, D(\mathbf{P}, \tilde{\mathbf{P}}) \tag{9}$$

where

$$^kU = \begin{cases} \emptyset, & if\ k = 1 \\ \{^1\mathbf{P}^*,\ ^2\mathbf{P}^*,\ ^3\mathbf{P}^*, ...,\ ^{k-1}\mathbf{P}^*\}, & otherwise \end{cases} \tag{10}$$

and

$$D(\mathbf{P}, \tilde{\mathbf{P}}) = \left\| \mathbf{P} - \tilde{\mathbf{P}} \right\| \tag{11}$$

The $D$ in Eq. 11 represents the Euclidean distance between the inputs $\mathbf{P}$ and $\tilde{\mathbf{P}}$, and $T_i^c$ is the complement of set $T_i$. Notice that if $\mathbf{P} \in T_i$, then $^k\mathbf{P}^*(\mathbf{P}) \in T_i^c$. For each image, we first determine the predicted class of the image. Then, we search for the nearest image (based on Euclidean distance) in the training set from the other class.

One challenge of using the single closest image as a baseline, instead of a black image, is that the explainability features might be affected by features from the baseline. To mitigate this issue, we took the weighted average of the attributions formed by the input and the 5 closest images from the other class. Each attribution was weighted inversely to the distance from the corresponding baseline image to the current input. The result is the Weighted Top 5 Closest Baseline (WT5CB) attribution. The following equations show the calculation for the IG case, but the form would be the same for all methods.

$$_{WT5CB_{IG}}\mathbf{C}(\mathbf{P}) = \sum_{k=1}^{5} {}_{IG}\mathbf{C}(\mathbf{P},\ ^k\mathbf{P}^*(\mathbf{P}))\beta_k \tag{12}$$

where

$$\beta_k = \frac{d^{-1}(\mathbf{P},\ ^k\mathbf{P}^*(\mathbf{P}))}{\sum_{k=1}^{5} d^{-1}(\mathbf{P},\ ^k\mathbf{P}^*(\mathbf{P}))} \tag{13}$$

## III. SOM AS GLOBAL EXPLAINABILITY TECHNIQUE

A Self-Organizing Map (SOM) [8] is a type of artificial neural network that is used for unsupervised clustering. The primary goal of an SOM is to produce a low-dimensional (typically two-dimensional) representation of high-dimensional data, while preserving the topological properties of the input space. This means that similar data points in the high-dimensional space are mapped to nearby locations in the low-dimensional topology of the SOM.

The SOM consists of a grid of neurons, each of which is associated with a weight vector of the same dimension as the input data. When an input vector is presented to the SOM, the algorithm identifies the neuron whose weight vector is most similar to the input vector. This winner neuron and its neighboring neurons' weights are then adjusted to become more like the input vector. This process is repeated for many input vectors, allowing the SOM to self-organize by grouping similar input vectors together into clusters and preserving the topological properties of the input space. The key feature of the SOM is its ability to cluster similar data points together and to visualize a high dimensional input space in two dimensions.

We will use the SOM to cluster input images from the full training and test sets of a trained DL classifier. Each of the resulting SOM clusters will contain similar images, and the images in neighboring clusters will be similar to each other. We can then identify: how the trained DL classifier responds to images across the SOM topology, where consistent errors occur, how features of images (like contrast and density) vary across the SOM topology, how DL classification varies across the SOM topology, etc. The SOM will give us a type of microscope or X-ray with which to examine the DL classifier operation. Our method employs several visualization tools to identify what the classifier model is focusing on to make decisions at a global level.

## IV. EXPERIMENT SETUP

Most explainability research in the image processing area has concentrated on object detection, object classification and image segmentation. Much less work has been done in explaining DL models for quality assessment. We believe that our approach is well-suited to this task. To demonstrate this, we need a suitable benchmark problem. We describe such a problem in this section.

Focused Assessment with Sonography in Trauma (FAST) ultrasounds are an important tool for rapid noninvasive evaluation. It is especially important for the identification of the presence of free fluid in the abdomen of traumatically injured patients. Quality assurance (QA) is a central process for all emergency department ultrasound activities and must be performed on all FAST exams. However, QA can take a significant amount of expert time and can be cost prohibitive. The authors in [11] report a DL model that automatically classifies the quality of FAST exams with 100% accuracy. However, they were not able to explain how the DL model was able to achieve this performance. Without a reasonable explanation, the model lacks full confidence from users.

We were able to obtain the DL models and the associated training and testing datasets from [11]. In the following sections we will demonstrate how to use our local and global explainability tools to identify the image features used by the DL models to accurately classify FAST exam quality.

The main objective of [11] is to distinguish between poor quality and good quality FAST exams. The dataset consists of 441 FAST exams, encompassing a total of 3,161 videos and 525,000 frames. The classification of the exams is based on individual frames extracted from these videos. For training, all videos are given the same label as the exam they came from, and all frames are given the same label as the video they came from. During inference, if half or more of the frames are assigned poor quality, the video is assigned poor quality, and if half of the videos are assigned poor quality, the exam is assigned poor quality.

The DL model was a convolution network, pretrained as an autoencoder and then fine-tuned for FAST quality classification with a two-layer classifier appended to the pre-trained encoder. (See [11] for more detail on the network architecture, training and inference.)

To train the SOM, we made two adjustments to the input data. Since training the SOM requires significant memory, we reduced the size of the dataset. We took only 20 randomly selected frames from each video in the dataset. This reduced the dataset to only 62,128 frames. Moreover, instead of using the frames themselves as inputs to the SOM, we used the encoder outputs. The original frames were 512x352 (180,224 pixels), and the encoder outputs were 3x64x44 (2,816 pixels).

## V. RESULTS

In this section, we cover the results of the proposed local and global explainability techniques. First, we compare the new WT5CB method with the original DeepLIFT and IG methods.

### A. Local Analysis of the Trained Network

*1) Comparison of WT5BC :* Figure 1, compares our WT5CB method with the original DeepLIFT and IG methods on one typical frame. One consistent feature across all the techniques is their emphasis on areas with distinct contrasts in the image. A prime example of this is the area (280:350, 175:225) in Figure 1. Each method seems to highlight this region, reinforcing the importance of such contrasting areas. Though all methods primarily focus on areas of contrast, the precision and the amount of noise vary between them. Both the DL Weighted Top 5 Closest Baseline (DL WT5CB) and the IG Weighted Top 5 Closest Baseline (IG WT5CB) methods outperform the original DeepLIFT and IG methods (black baseline [BB]). Notably, these methods place precise emphasis on areas of contrast. Among them, the DL WT5CB method is found to be the most precise.

To showcase the precision of the DL WT5CB method, we conducted a comparative analysis with the DL BB method. In Figure 2, a distinct area taken from Figure 1, specifically denoted as (100:300, 250:350), is highlighted. Upon close observation of this region, the superior precision of the DL WT5CB method becomes readily apparent when compared to the DL BB approach. We found the increased precision consistently across a variety of frames that we tested.

*2) Assessment using the DL WT5CB Method:* In Figures 3 and 4, we plotted the results of two US frames extracted from the same video, spaced one second apart. The frame depicted in Figure 4 produces a network output of 0.24, indicating good quality, while the frame in Figure 3 produces an output of 0.68, indicating poor quality. (A network output of 1 represents poor quality, and a network output of 0 indicates good quality.) Despite the striking similarities of the two frames — their origin from the same video and their temporal proximity — and the attribution methods pointing out similar regions of interest, they were classified differently. This leads to the question: What differences in these two images prompt such varied classifications?

The answer may lie in the image's sharpness. For example, in Figure 4, sharp contrasts are distinctly visible around coordinates (310, 280), (370, 80), and (450, 285). These regions are underscored by the attributions across both figures, suggesting that such features heavily influence the network's

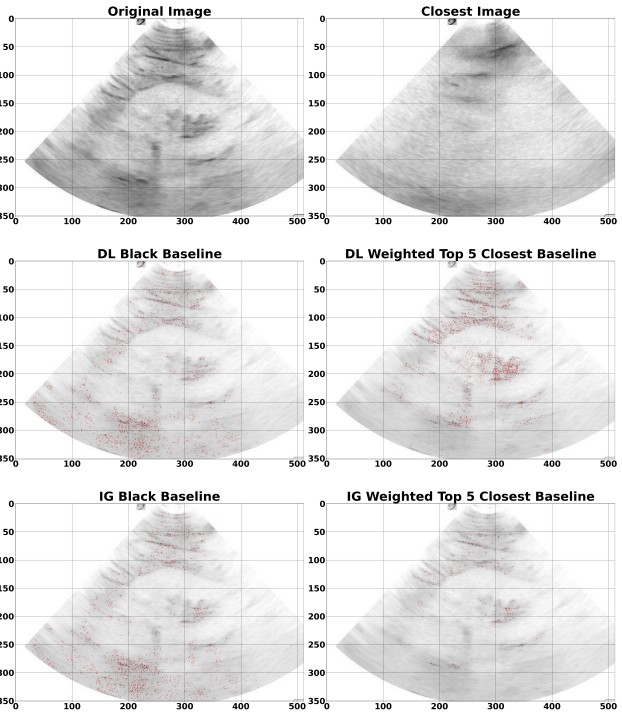

Fig. 1. Explainability results for a typical FAST frame – Red pixels indicate high attribution levels – most important for the decision

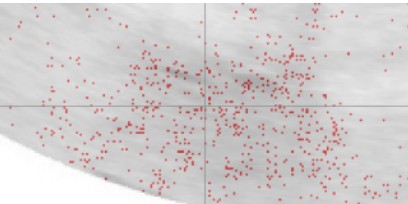

(a) Zoom of DL BB from area at (100:300, 250:350) in Figure 1

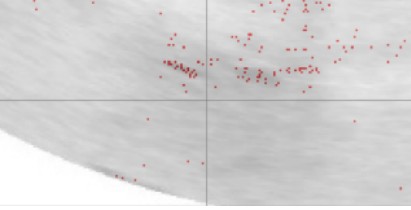

(b) Zoom of DL WT5CB from area at (100:300, 250:350) in Figure 1

Fig. 2. Zoomed comparison of DL Black Baseline (BB) and DL Weighted Top 5 Closest Baseline (WT5CB) for the FAST frame in Figure 1

decision-making process. As a result, the network classifies the frame in Figure 4 as being of good quality. However, in the same locations in Figure 3, the contrast appears fuzzy, indicating that the frame is of poor quality. It appears that image sharpness is a key characteristic valued by the network.

In Figures 5 through 7, we present a side-by-side comparison that specifically emphasizes the differences between

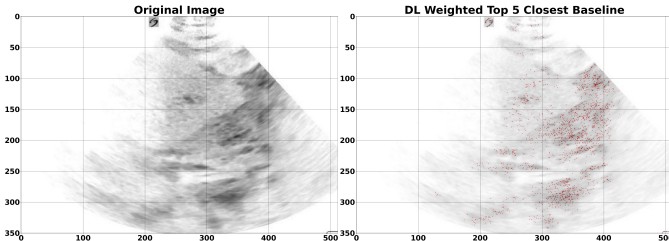

Fig. 3. Example of an image classified as poor quality and its DL attributions.

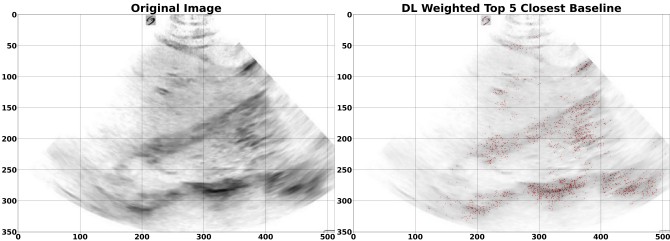

Fig. 4. Example of an image classified as good quality and its DL attributions.

the three regions highlighted in the preceding discussion. This comparative layout provides a clearer, visual understanding of how quality variations manifest in the two figures (Figure 4 and Figure 3). Notice that the subimages extracted from Figure 4, which the network classified as good quality, display distinct sharpness levels. In comparison, the corresponding subimages from Figure 3—which was classified as poor quality—display a noticeable fuzziness. Such characteristics may be the factors that influenced the network's decision to classify the frame as of poor quality. These juxtapositions in Figures 5 through 7 offer tangible evidence of the disparities in quality between the two frames, underscoring the importance of sharpness in the evaluation of image quality. This will be reinforced when we consider the SOM analysis.

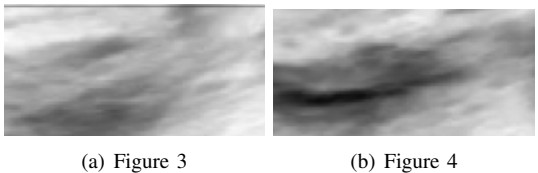

(a) Figure 3        (b) Figure 4

Fig. 5. Zoomed illustration of region (300:360,270:290) in Figures 3 and 4.

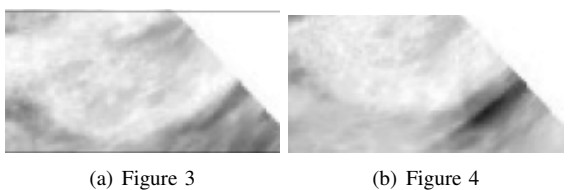

(a) Figure 3        (b) Figure 4

Fig. 6. Zoomed illustration of region (360:390,70:90) in Figures 3 and 4.

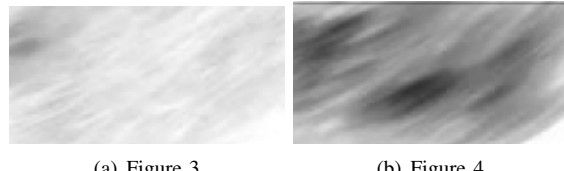

(a) Figure 3        (b) Figure 4

Fig. 7. Zoomed illustration of region (440:460,270:290) in Figures 3 and 4.

### B. Global Analysis of the Trained Network

So far, we analyzed classifier network performance using local explainability techniques. In this section, we explain the proposed novel approach to global explainability, employing the SOM as a global explainer. We utilize various visualization tools of the SOM to elucidate the observed patterns.

We trained 5x5, 7x7 and 10x10 hexagonal SOMs on the encoder outputs of 62,128 FAST frames. The encoder output dimension was 2,816. We found that the 7x7 SOM enabled the best analysis.

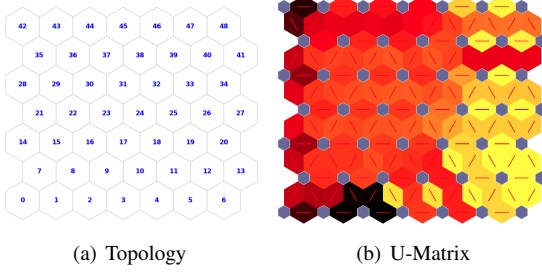

(a) Topology        (b) U-Matrix

Fig. 8. Topology and U-Matrix (indicating distances between clusters – darker color for greater distance) for the SOM trained on FAST frames.

The topology of the SOM is given in Figure 8(a). The units are numbered from left to right and from bottom to top. In the next sections, we will be referring to these numbers as cluster numbers.

Figure 8(b) displays a Unified Distance Matrix, commonly referred to as the U-matrix, which is utilized to illustrate the distances between adjacent weight vectors, or cluster centers. The small, dark gray hexagons symbolize the cluster centers, while the colored, elongated hexagons depict the distances between these centers. A darker shade denotes a greater distance. For instance, the significant distance between clusters 2 and 3 is represented by a dark color. Conversely, a lighter shade between clusters 6 and 13 suggests that these centers are relatively closer together. The U-matrix can help us identify super-clusters within the SOM. For example, clusters 6, 12. 13, 19 and 20 are all close to each other, which indicates that the frames within this set of clusters will be similar.

Figure 9(a) presents a Hit Histogram, with each hexagon symbolizing a cluster center. This histogram indicates the number of frames in each cluster. The hexagon size reflects the frame count, which is also indicated inside each hexagon.

The clusters vary a lot in size, however, Figure 9(b) shows that there is less variation in the cluster radii. The larger clusters are more densely packed.

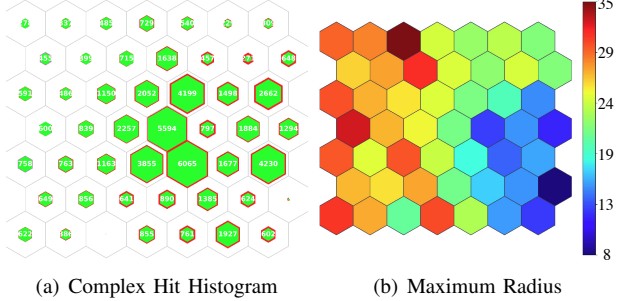

| (a) Complex Hit Histogram | (b) Maximum Radius |

Fig. 9. Complex hit histogram, indicating the number of frames in each cluster, and the maximum radius of each cluster, indicating the variation in the frames in each cluster.

Now that we have clustered the inputs to the CNN, we can look for patterns of network operation. How does the network classify various types of FAST frames? For example, Figure 10 illustrates the percentage of False Positive (FP) frames across the clusters (positive represents poor quality), highlighting that those in the Bottom Right (BR) section of the topology exhibit relatively higher FP rates. This observation indicates that frames of good quality in the Top Left (TL) section are classified more accurately than those in the BR section.

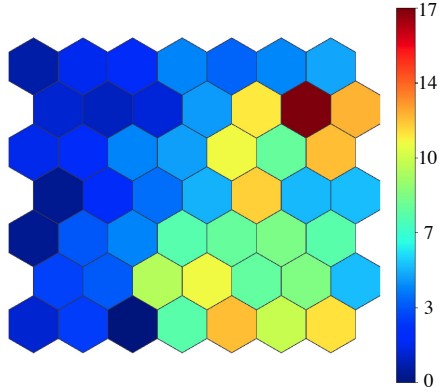

Fig. 10. Average False Positive (FP) percentage in each cluster.

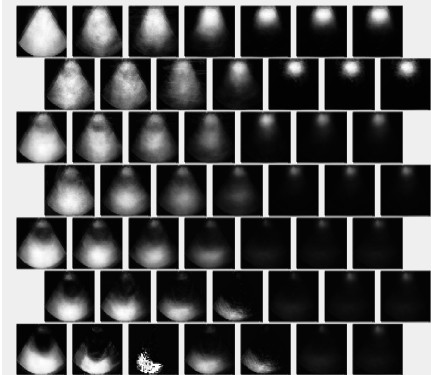

Fig. 11. Cluster centers (first feature map), indicating typical frame shape within each cluster. (See Figure 8 for cluster placements.)

In Figure 11, we show the cluster center for each cluster. (The figure displays the first feature map generated at the

Encoder's output.) These cluster centers are representative of typical frames in each cluster. Frames that are relatively dense are predominantly located in the TL part of Figure 11. For instance, clusters 21, 22, 28, 29, 35, 36, 42, and 43 feature dense frames. In contrast, clusters in the BR section, such as clusters 5, 6, 12, 13, 19, 20, 26 and 27, have relatively sparse frames.

Thus far we learned that clusters in the TL area have frames that are dense and contain a small number of FPs. Conversely, clusters in the BR section contain sparse frames, and there is a higher percentage of FPs.

More detail on network operation is shown in Figure 12, which, as well as FP, includes False Negative (FN), True Negative (TN) and True Positive (TP): TP in green, TN in blue, FN in yellow, and FP in red. Figure 10 showed that FP errors predominantly accumulate in the BR section. It's now observable that most clusters in the TL section are characterized as TNs, and this region is associated with cluster centers exhibiting dense frames, as shown in Figure 11. This could indicate that dense frames are generally classified by the network as good quality. (The dataset is unbalanced, with 94% of frames classified as good quality.)

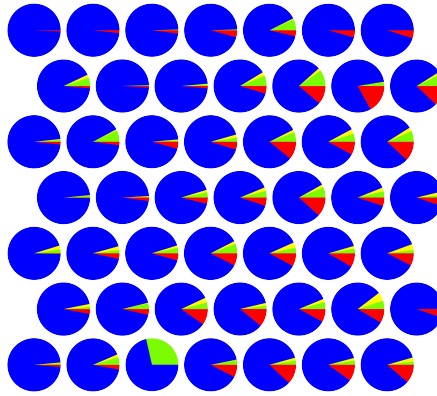

Fig. 12. Classification performance for each cluster: TP(g), FN(y), TN(b), FP(r)

Figure 13 reinforces the idea that the network classifies sparse frames as low quality. It shows the heat map for the average output in each cluster. Clusters in the TL area, predominantly consisting of dense frames, show a lower average output, indicative of good quality. In contrast, the BR section, characterized by sparser frames, exhibits a higher average output, signaling poor quality.

We further analyzed the distribution of outputs for each cluster using histograms, as shown in Figure 14. The bins in the histograms represent output values ranging from 0 (good quality) to 1 (poor quality), in increments of 0.1. We observed a noticeable skew towards the left (closer to zero) in the outputs in the TL section, indicating that dense frames are more likely to be good quality. The confidence of the network to classify frames in the TL section as good quality is high. As we move to the BR section, the output begins to shift towards the center, which indicates a lower confidence in determining

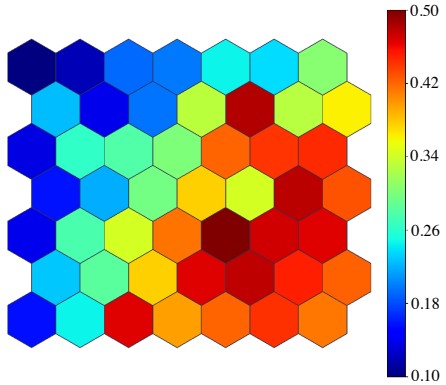

Fig. 13. Average classification network output for each cluster.

both good and poor quality frames. This figure also shows a higher variation in terms of output as we go farther from the TL.

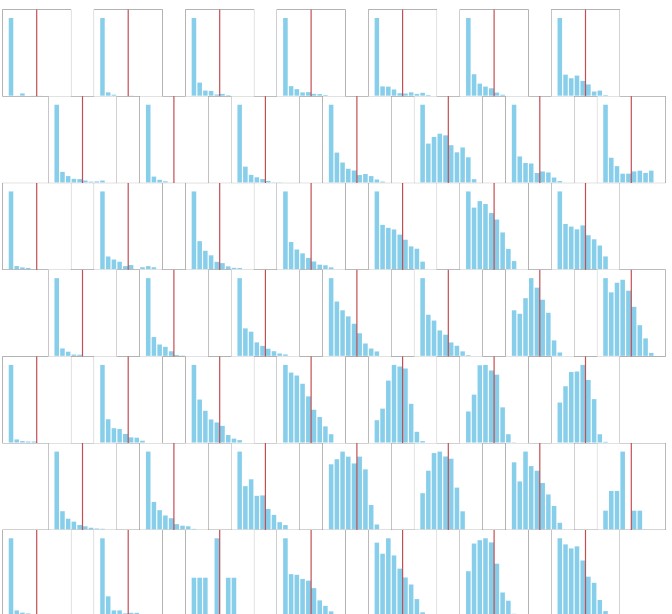

Fig. 14. Classification network output histogram by cluster. Red line represents 0.5 – smaller is good quality, larger is poor quality. (See Figure 8 for cluster placements.)

In order to have a quantitive measure of sparsity, for each image, we counted the number of pixels with a value greater than or equal to 30 on a 0 to 255 scale. We then calculated the ratio of this number to the total number of pixels (352*512). A higher ratio indicates greater density. In Figure 15, we plotted this ratio against the network output and also included a linear regression line. The negative slope of the regression line shows that sparse frames tend to have a higher output (poor quality), while denser images are likely to be good quality.

Our SOM analysis shows that, globally, our classifier considers sparsity as an indicator of frame quality. In the TL section of the SOM, this is very clear, as the network confidently classifies very dense frames as good quality. For the sparser frames that appear more often as we move toward

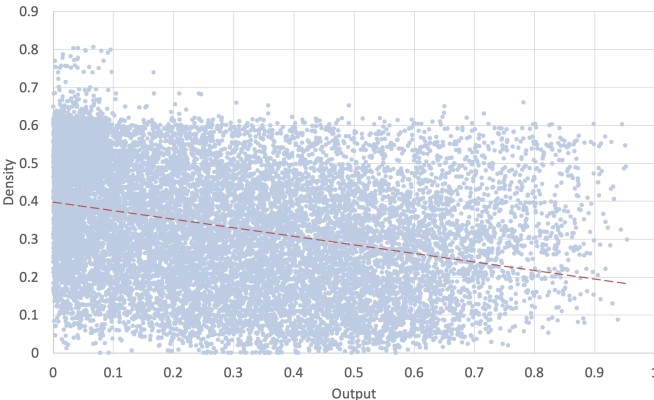

Fig. 15. Density vs classification network output for all FAST frames (correlation coefficient = -0.442).

the BR of the SOM, there is more variation in network output. This raises the question: What other factors does the network consider when assessing the quality of a frame? Recall that, in Section V-A2, we discussed how sharpness affected the network's decision-making process using local explainability. Here, we will explore sharpness in a global sense.

To quantitatively measure sharpness (approximately), we first applied the Laplacian filter to the frames. The Laplacian filter produces a new image that represents the second derivative of the original image, which is used to find regions of rapid intensity change in frames. After applying the Laplacian operator, the contrast of the image can be quantified by calculating the standard deviation of the resulting image. The standard deviation measures the amount of variation or dispersion of a set of values. In the context of the Laplacian-applied image, a higher standard deviation (STD) indicates a greater presence of high-contrast regions.

For each frame within each cluster, we calculated the Laplacian STD and plotted it against the network output in Figure 16. The horizontal axis represents the network output, ranging from 0 to 1, while the vertical axis corresponds to the Laplacian STD. We also plotted a linear regression line to show the trend. This figure shows that there is an inverse relationship between the network output and the Laplacian STD in almost every cluster, suggesting that higher contrast or sharpness (indicated by a higher Laplacian STD) increases the likelihood of the network classifying the frame as good quality – reflected by a lower network output. (The few clusters with positive slope contained outlier images.)

### C. Summary

By combining the global analysis of the SOM with the local explainability techniques, we were able to show that the DL model was using two different characteristics of the FAST frames to identify quality: the sharpness of the images and the density of the images. It would not have been possible with standard local explainability methods to identify density as a key characteristic, since density is a global feature of an image.

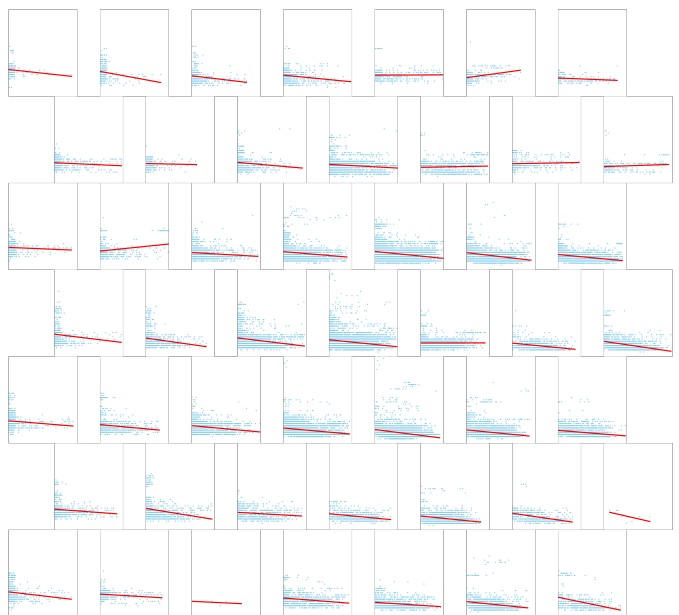

Fig. 16. Sharpness vs output by cluster (See Figure 8 for cluster placements.)

## VI. Conclusions and Future Work

This research introduced innovative enhancements to existing local explainability methods and utilized the Self-Organizing Map (SOM) as a global explainability method, providing a more profound insight into how deep neural networks reach decisions. Our methods are especially well-suited to explaining DL models that assess image quality.

Local explainability techniques, such as DeepLIFT and Integrated Gradient, are commonly used to understand neural network decisions at a detailed level. Typically, these methods use a single baseline image, often entirely black, for their computations. However, this study introduces a novel approach by utilizing multiple baseline images from the training dataset that are closest to the input image but belong to different categories. This method forces the baseline images to be on the opposite side of the decision boundary from the input image, resulting in more precise and informative explanations compared to conventional methods.

Additionally, the study presents a fresh approach to global explainability, aiming to provide a broader understanding of the neural network's decision process. The Self-Organizing Map (SOM) is employed as a global explainer, facilitating the clustering of inputs and the visualization of results in a two-dimensional grid. This approach allows for the identification of data patterns and offers insights into areas where the neural network may have misclassified inputs.

To validate these methods, the study applied them to a deep learning model for the quality classification of FAST ultrasound images. The results showed that the classification network could accurately assess the quality of ultrasound images based on two main characteristics: image sharpness and image density. These findings are results of the combined local and global explainability techniques. These findings not only enhance our comprehension of how neural networks interpret medical imaging data but also have implications for enhancing the accuracy and reliability of such systems in practical applications.

We have not yet brought this technology to a clinical setting or considered other imaging modalities. These are promising avenues that could be pursued in future work. For example, when the deep model makes a classification of a FAST exam, it could display several frames that contributed to the decision and include overlays on the frames of pixels with maximum attribution. It could also display density and sharpness values for each frame, and indicate which SOM clusters the frames fell into. Of course, there would need to be extensive tests with clinicians to determine the best variables and formats to use in displays, since explanations are only useful if they clarify the decisions, and this is a subjective question. It also seems likely that the analyses described in this paper could be applied to other imaging modalities, especially those assessing quality.

## Acknowledgment

We would like to thank Drs. Dustin Morrow and John Cull of Prisma Health and Dr. Hudson Smith of Clemson University for the data and problem definition.

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
