# OpenReview forum: "Explainability for Quality Classifiers in Ultrasound"
_IEEE.org/EMBS/BHI/2024/Conference — IEEE BHI'24_

### Official Review · Reviewer_4dEA · 2024-08-01
**Explainability for Quality Classifiers in Ultrasound**

**Overall Rating:** 7
**Confidence:** 5

**Other Quality Metrics:**

Clarity of writing: Good

Clinical significance: Fair

Methodological novelty: Good

Experiments and results: Good

**Questions For The Authors:**

None

**Strengths:**

This study presents a novel approach to global explainability, aiming to provide a broader understanding of the neural network’s decision process

**Summary Of The Paper:**

This study introduces a novelty to two types of local explainability techniques by utilizing
the Self-Organizing Map (SOM) and aims to achieve a combination of local and global explainability.

This approach allows for the identification of data patterns and offers insights into areas where the neural network may have misclassified inputs. The results showed that the network classification could accurately evaluate the quality of ultrasound images based on two fundamental characteristics: image sharpness and image density. These findings are results of the combined local and global explainability techniques and they also have implications for enhancing the accuracy and reliability of such systems in practical applications.

**Weaknesses:**

None

---

### Official Review · Reviewer_HAyt · 2024-08-08
**Explainability for Quality Classifiers in Ultrasound**

**Overall Rating:** 8
**Confidence:** 5

**Other Quality Metrics:**

(a) Clarity of writing; excellent
 (b) Clinical Significance; excellent
 (c) Methodological Novelty; Great
 (d) Experiments and Results: Great

**Questions For The Authors:**

Congratulations on your innovative work in enhancing explainability for healthcare quality classification models. Your approach is both timely and impactful.
Clarification on Clinical Application: How do you envision this solution being used in a real clinical scenario? Specifically, how would clinicians interact with the explainability framework, and what training or tools would they need to interpret the model outputs effectively?
Generalization Across Datasets: Have you considered testing your framework on other datasets? How do you anticipate it will perform across different types of healthcare data?

**Strengths:**

Novelty: The approach to enhancing explainability in healthcare quality classification is innovative and addresses a critical gap in the current research.
Comprehensive Explanation: The concepts, problem, and solution are well-explained, making the work accessible and understandable.
Practical Relevance: The work has significant potential for real-world applications in clinical settings, improving trust in machine learning models among healthcare professionals.

**Summary Of The Paper:**

The paper presents a novel approach to enhancing the explainability of machine learning models used in the classification of healthcare quality data.
The authors have developed a framework that leverages a large dataset for training a machine learning model, focusing on improving transparency and interpretability.
The study aims to address the challenges in understanding the decision-making process of complex models in healthcare, particularly in ensuring that predictions can be trusted and validated by clinicians.
The work is grounded in solid theoretical concepts, well-defined problems, and a clearly articulated solution.
The experiments are comprehensive, involving a large dataset, and the results demonstrate the efficacy of the proposed approach.

**Weaknesses:**

Clinical Integration: The paper discusses explainability but could provide more details on how the proposed solution would be integrated into existing clinical workflows. This would help in assessing the practical utility of the approach.
Complexity of Implementation: The paper could explore the potential challenges in implementing this solution in real-world scenarios, including the computational resources required and the training needed for clinicians to interpret the model outputs effectively.

---

### Official Review · Reviewer_YxqX · 2024-08-12
**Improving explainability of a deep learning classifier demonstrated in ultrasound imaging**

**Overall Rating:** 6
**Confidence:** 4

**Other Quality Metrics:**

(a) Clarity of writing; fair (there is room to improve the quality of writing to make it a formal scientific and archival manuscript. The figures require substantial adjustment)

 (b) Clinical Significance; good (ultrasound medal images are used as an example of the practicality of the proposed machine learning application)

 (c) Methodological Novelty; good (a convincing way of improving the transparency in machine learning applications is proposed: baseline selection near the current output but on the opposite side of the decision boundary)

(d) Experiments and Results; good (systematic data analysis, visualization and interpretation)

**Questions For The Authors:**

In section IV, please fix ‘The authors in in [20] report’

Please revise the caption of Fig. 1. What does high-levels mean?

Please introduce the abbreviations BB and WT5CB in the caption of Fig. 2. If possible, add a scalebar to the subfigures of Fig. 2. Please indicate the taken distinct areas within Fig 1.

Please indicate in Figs. 3-4 where the selected regions shown in Figs. 5-7 correspond to.

In Fig. 9(b) 10, and 13, please make the font-size of colorbar the same as that of caption and add  labels for the colorbars.

Page 6 on the right column, ‘We’ve noted before’ please rewrite it in the formal form.

Combined global and local explainability techniques are shown to enhance the comprehension of how neural networks interpret ultrasound medical images with some implications on the accuracy and reliability in practical applications. Please discuss whether your proposed method can be applied on other medical imaging modalities such as laser speckle perfusion imaging in addressing motion artifacts (automatically choosing a proper reference image/region for motion tracking and image registration, and using regression analysis to improve the quality of a physical model for motion artifact prediction such as the one used in https://doi.org/10.1117/1.jbo.28.4.046005).

**Strengths:**

Introducing a different approach to enhance the explainability of a DL model, namely, choosing the baseline data close to the current input but on the other side of the boundary of decision making.

application of explainability in quality assessment in medical imaging rather than for conventional segmentation and object detection.

**Summary Of The Paper:**

The paper has used self-organizing-map (SOM) as a method to address the explainability of a deep learning (DL) trained in a supervised way demonstrated in focused assessment with sonography in trauma ultrasound.

**Weaknesses:**

The title sounds too general and not specific to the study of this paper. It could be revised for instance: Improving explainability of classification based on modified baseline selection demonstrated in ultrasound medical imaging

The abstract lacks a statement on background and connection to clinics (clinical problem).

Figure captions are abstract and require additional information to make them independent and informative.

The message of Fig. 11 is not clear. This can be addressed by adding titles or numbers to the subfigures. The same comment hold for Fig. 14.

Fig. 15 requires a major modification. All font-sizes should be the same. No bold-style for the x- and y-labels. Caption requires a complete and concise description of the shown data. The goodness of fit and basic parameters of the fitted line should be shown, as well.

---

### Decision · Program_Chairs · 2024-09-23

Accept